# Effect of Body Mass Index on Postoperative Nausea and Vomiting: Propensity Analysis

**DOI:** 10.3390/jcm9061612

**Published:** 2020-05-26

**Authors:** Jong Ho Kim, Mingi Hong, Young Joon Kim, Ho Seok Lee, Young Suk Kwon, Jae Jun Lee

**Affiliations:** 1Department of Anesthesiology and Pain Medicine, Chuncheon Sacred Heart Hospital, College of Medicine, Hallym University, Chuncheon 24253, Korea; poik99@hallym.or.kr (J.H.K.); joonnjin@hallym.or.kr (Y.J.K.); guacamole@hallym.or.kr (H.S.L.); 2Institute of New Frontier Research Team, Hallym University, Chuncheon 24253, Korea; buriminki@naver.com

**Keywords:** body mass index, obesity, overweight, postoperative nausea and vomiting

## Abstract

The impact of body mass index (BMI) on postoperative nausea and vomiting (PONV) is controversial, and few studies have focused on their relationship. We investigated the effects of BMI on PONV, taking into account other PONV risk factors. We analyzed adults over the age of 18 years who received general anesthesia between 2015 and 2019, using propensity score matching. Before propensity score matching, odds ratios (ORs) for PONV were lower for overweight (OR, 0.91; 95% confidence interval (CI), 0.87–0.96; *p* < 0.0001) or obese patients (OR, 0.77; 95% CI, 0.71–0.84; *p* < 0.0001) than for normal-BMI patients. After matching, the ORs for PONV of overweight (OR, 0.89; 95% CI, 0.80–0.98; *p* = 0.016) and obese patients (OR, 0.71; 95% CI, 0.63–0.79; *p* < 0.0001) were low. However, the ORs of underweight patients did not differ from those of normal-BMI patients, irrespective of matching. Therefore, the incidence of PONV may be lower among adults with a higher-than-normal BMI.

## 1. Introduction

Postoperative nausea and vomiting (PONV) are common complications of anesthesia [1] that occur in 30% of hospitalized patients; moreover, their incidences are significantly greater in high-risk patients [2]. In previous studies, vomiting was the most undesirable outcome of patients after surgery, and nausea was the fourth most undesirable outcome [3]. PONV is usually non-fatal [4] but, in rare cases, can cause serious pathological conditions such as dehydration, electrolyte imbalance, suture tension and dislocation, venous hypertension and bleeding, aspiration of gastric contents, esophageal rupture, and life-threatening airway damage [5,6,7].

Several studies have identified factors that alleviate or exacerbate PONV [2,5,6,8,9,10]. However, whether obesity is a risk factor for PONV is controversial. A relatively high body mass index (BMI) may promote PONV [8,11], but the opposite effect has also been reported [12,13,14]. Others have found that BMI is not a risk factor for PONV [15,16].

Many previous PONV studies were retrospective observational studies [12,13,17,18,19]. As observational studies conduct research in specific groups without random assignment, they are slightly more realistic representations of the clinical situation, and statistical controls can be applied for specific confounding variables. However, observational studies cannot avoid selection bias. It is fundamentally impossible to deduce the cause of a phenomenon in studies that are not based on random assignment. Matching methods are generally used in observational studies to reduce selection bias [20]. Propensity score matching (PSM) is used not only in observational studies but also in retrospective research where it is difficult to apply random assignment [21].

We investigated the effects of BMI on PONV, taking into account other risk factors for PONV. The validity of observational studies on the effects of BMI can be reduced by selection bias and confounders, so we performed propensity score matching.

## 2. Patients and Methods

### 2.1. Patients

The data were obtained from the Clinical Data Warehouse of Hallym University Medical Center, and the study was approved by the Clinical Research Ethics Committee of Chuncheon Sacred Heart Hospital, Hallym University (Institutional Review Board no. 2020-03-011). The patients were at least 18 years of age and had undergone surgery under general anesthesia at Hallym University Medical Center between January 2015 and December 2019. The exclusion criteria were as follows: patients who received repeat surgery within 24 h after anesthesia, patients who were unconscious after surgery, patients who underwent ventilator therapy after anesthesia, patients with nausea or vomiting before anesthesia, and patients with data missing from their medical records.

### 2.2. BMI, PONV, and Covariates

BMI is defined as weight (in kilograms) divided by the square of the height (in meters). In this study, patients with BMI <18.5, 18.5–25, 25.1–30, or >30 were considered underweight, normal weight, overweight, or obese, respectively. PONV was defined as the occurrence of nausea or vomiting within 24 h after anesthesia as recorded in medical records. This study did not consider nausea or vomiting events as separate outcomes because they have identical risk factors and predictors [17,22,23,24,25,26,27], postoperative nausea may be considered a symptom for potential vomiting [28], and the accompanying occurrence of nausea and vomiting is more than the individual occurrences of nausea and vomiting [6]. Covariates included age; sex; height; operation duration; American Society of Anesthesiologists physical status; use of N_2_O during anesthesia maintenance; patient-controlled analgesia after surgery; history of diabetes, gastroesophageal reflux disease, or smoking; use of antiemetics, opioids, proton-pump inhibitors, or antibiotics; use of a Levin tube during and after surgery; laparoscopic surgery; and transfusion during surgery.

### 2.3. Statistics

Continuous data are presented as means and standard deviations and categorical data as frequencies and percentages. For continuous data, *t*-tests were performed to compare between patients with and without PONV. Categorical data were analyzed by chi-squared test. First, the odds ratio (OR), with 95% confidence interval (CI), for occurrence of PONV within 24 h after surgery was calculated for each variable by logistic regression. The odds ratio (OR) is a measure of the association between an exposure and an outcome [29], and the OR in this study represented the likelihood of PONV given a particular exposure (underweight, overweight, obese) compared to the likelihood of PONV in the absence of this exposure. Next, the adjusted OR for PONV according to BMI category was determined. Finally, the ORs adjusted for all variables and for those variables selected by backward elimination were calculated.

As patients cannot be randomized based on BMI category, confounding and selection bias were accounted for by developing propensity scores for the BMI categories. The rationale for, and method of, using propensity scores for the proposed causal exposure variables have been described elsewhere [30,31]. We performed three rounds of propensity score matching: normal BMI matched with underweight, normal BMI matched with overweight, and normal BMI matched with obese patients. Anaconda (Python version 3.7, https://www.anaconda.com; Anaconda Inc., Austin, TX, USA) and pymatch (version 0.3.4; https://github.com/benmiroglio/pymatch) were used for propensity score matching. The propensity scores were 0.26–0.87, 0.30–0.68, and 0.21–0.83 for matched normal and underweight, normal and overweight, and normal and obese patients, respectively. All matched records had scores within 0.0001 of each other, and all matching ratios were 1:1. We matched 4063 underweight patients with 4063 normal-BMI patients, 9872 overweight patients with 9872 normal-BMI patients, and 7678 obese patients with 7678 normal-BMI patients.

The ORs for PONV in underweight, overweight, and obese patients were compared with that of normal-BMI patients. In each analysis, covariates and propensity scores were used to calculate the adjusted OR. All *p*-values were two-sided, and a *p*-value < 0.05 was considered indicative of statistical significance. SPSS software (version 24.0; IBM Corp., Armonk, NY, USA) was used for the statistical analyses. The dataset used is provided in Appendix A.

## 3. Results

### Patient Characteristics

Between January 2015 and December 2019, a total of 113,881 patients underwent general anesthesia in Hallym University Medical Center. After excluding 10,320 patients, 103,561 were included in the study (Figure 1). PONV occurred within 24 h after surgery in 10,404 of these patients; their baseline characteristics are listed in Table 1. The adjusted and unadjusted ORs of all variables for developing PONV are listed in Table 2. The ORs for PONV were higher in overweight (unadjusted OR, 0.85 (95% CI, 0.81–0.89); fully adjusted OR, 0.91 (95% CI, 0.87–096)) and obese (unadjusted OR, 0.79 (95% CI, 0.73–0.86); fully adjusted OR, 0.77 (95% CI, 0.71–084)) patients compared with normal-BMI patients.

Between normal-BMI and underweight patients, there were significant differences in 10 variables before matching and no significant difference in any variable after matching (Table 3). Between normal-BMI and overweight patients, there were significant differences in 14 variables before matching and a significant difference in 1 variable (diabetes, *p* = 0.008) after matching (Table 4). Between normal-BMI and obese patients, there were significant differences in nine variables before matching and no significant difference in any variable after matching (Table 5).

Figure 2 shows the ORs for development of PONV in underweight, overweight, and obese patients. The unadjusted and adjusted ORs for PONV did not differ significantly between underweight and normal-BMI patients. The unadjusted (*p* = 0.046) and adjusted (*p* = 0.015–0.016) ORs for PONV were lower in overweight patients than in normal-BMI patients. Moreover, the unadjusted (*p* < 0.0001) and adjusted (*p* < 0.0001) ORs for PONV were lower in overweight patients than in normal-BMI patients.

## 4. Discussion

We estimated the effects of being underweight, overweight, and obese based on BMI on the incidence of PONV among 113,881 patients who underwent general anesthesia during a five-year period. The ORs for PONV were lower for overweight and obese patients compared with normal-BMI patients. However, there was no difference in the ORs for PONV between underweight patients and normal-BMI patients. After propensity score matching, the results were essentially unchanged: overweight and obese patients exhibited lower ORs for PONV compared with normal-BMI patients.

Most anesthesiologists consider that an increase in BMI increases the incidence of PONV [11]. The reason may be explained as follows. Longer anesthesia time is associated with longer reduction time of the anesthetic in obese patients [32]. The action time of the anesthetic is increased when a hydrophilic anesthetic is administered based on total body weight [33]. Obese patients have high incidence rates of diabetes [34] and gastroesophageal reflux disease [35] as well as elevated abdominal pressure [36]. Bellville et al. reported that obese patients have a significantly higher incidence of nausea and vomiting compared with those who are thin, because they have a large fat reservoir to saturate [37]; others have confirmed this positive correlation [8,38,39,40,41]. However, the rate of recovery from anesthesia also depends on the time at which the anesthetic is administered [42]. Obesity is assumed to delay recovery, especially from prolonged anesthesia, but it is unclear how changes associated with obesity (such as fat distribution and increased mass) affect the pharmacokinetics of inhaled anesthetics during anesthesia. Obesity reportedly does not affect awakening time after anesthetic administration [43]. Obesity is associated with an increased adipose tissue mass, which accounts for 20–40% of the excess weight in obese individuals. Non-fat mass includes well-dispersed tissues, such as organ and muscle mass [44,45], in which anesthetics areabsorbed and diffuse quickly. Fat adjacent to well-perfused tissues rapidly absorbs anesthetics, but transfer of anesthetics between these tissues, such as subcutaneous fat or skeletal muscle fat, increases over 3 h [43]. N_2_O has a low fat/blood partition coefficient and is removed from the body rapidly [42].

Although a relatively high BMI has been reported to be associated with the development of PONV, our findings showed that an increased BMI reduced the incidence of PONV. However, the mechanism underlying this inverse association is unknown. Drug elimination is generally more rapid in obese patients than in non-obese patients [46]. The clearance of some volatile anesthetics, prednisolone, and some benzodiazepines is sometimes faster in obese patients [47]. Opioids are widely used as adjuvants for anesthesia and are risk factors for PONV [42,48]. However, opioids administered during surgery have little direct effect on PONV, irrespective of the opioid in question [49]. Most opioids are biotransformed primarily in the liver, via metabolism by the cytochrome *p* (CYP) system, conjugated in the liver, or both [42,50]. Obesity causes anatomical, physiological, and metabolic changes that can affect the pharmacokinetics of a drug (including its absorption, distribution, metabolism, and excretion) [51,52,53,54,55]. However, there is no correlation between CYP activity and BMI [50]. The pharmacokinetics of propofol, which is frequently used for anesthesia induction, are not affected by obesity [42]; moreover, propofol has antiemetic activity [56].

Kranke et al. reported that a relatively high BMI was not a risk factor for PONV [15]. In contrast, in this study, higher BMI was associated with a decreased incidence of PONV. A systematic review with meta-analysis is considered the best means of providing definitive answers to research questions. However, such analysis has flaws, such as the location and design of the studies analyzed, heterogeneity, loss of data, improper subgroup analysis, conflict with new experimental data, and publication overlap [57].

No mechanism for the reduced incidence of PONV in patients with higher-than-normal BMI has yet been reported, but histamine, a neurotransmitter that stimulates various pathways and receptors that mediate nausea and vomiting [58], may play a role. Leptin is an adipocyte-derived hormone that suppresses appetite, increases energy expenditure and activation of the histaminergic system in the hypothalamus, and controls body weight [59]. The amount of body fat is directly correlated with circulating leptin levels, and serum leptin level is elevated in obese individuals and drops during weight loss or fasting. The amount of reduction during the fasting period is significantly reduced compared to individuals of normal weight [60]. The decrease in activity of the histamine activation system may be greater in subjects with higher-than-normal BMI during perioperative fasting. Dopamine receptors, especially D2 and D3, have been shown to play important roles in inducing nausea and vomiting. This mechanism blocks adenylate cyclase, reducing the amount of cAMP in neurons in the solitary nucleus and area postrema [58,61]. Because dopamine level and global activation of dopaminergic systems are increased by stress [62], PONV can be increase after severe stress such as surgery. However, obese patients havefewer dopamine receptors than those with normal BMI [63], so patients with obesity may have less effect of dopamine for PONV than normal-BMI or underweight patients. Anticholinergics are one of the antiemetic agents. They block muscarinic receptors in the cerebral cortex and pons, causing an antiemetic effect [61]. In reversal of muscle relaxation, anticholinergics areadministered concurrently with cholinesterase inhibitors to reduce the side effects of cholinesterase inhibitors. In obese patients, more anticholinergic drugs are given to reverse muscle relaxation, which may be associated with a lower incidence of PONV in obese patients.

Studies of the effect of type of surgery on the incidence of PONV have yielded conflicting results [64]. A systemic review of PONV suggested that laparoscopic surgery is associated with increased risk of PONV [21]. In this study, we included laparoscopic surgery as a covariate for the incidence of PONV. Strabismus surgery [27,65,66], adenotonsillectomy [67], and inguinal scrotal and penile procedures [68] have been reported as risk factors for PONV. However, we did not analyze the surgeries separately because these previous studies were mostly performed in pediatric patients and thus the results cannot be directly compared to our adult cohort.

Our results support and extend previous findings in several important respects. First, we demonstrated that a relatively high BMI is associated with a reduction in the incidence of PONV, as reported by others [12,13,14]. Second, because we focused on BMI, we were able to account for the effect of BMI category (underweight, normal BMI, overweight, and obese) on the incidence of PONV. Most PONV studies did not focus on BMI [1,3,5,6,8,11,12,13,14]. However, this study also had several limitations. First, classification of BMI cannot be randomized. Although the use of observational studies to assess effectiveness is controversial [69], rigorous observational studies may not yield misleading or biased results [70,71]. Moreover, to prevent selection bias and confounding, we performed propensity score matching with tighter adjustments than are typically used in a general multivariate analysis [30]. Nevertheless, observational studies can only partially control the element that is actually measured, and it is difficult to control elements that cannot be measured [72]. Second, although we used BMI to classify patients as underweight, overweight, and obese, BMI does not differentiate between lean body mass and fat mass [73]. In addition, BMI does not reflect the distribution of adipose tissue, which affects the absorption and distribution of a drug [43,73]. However, BMI is a convenient rule of thumb for classifying certain categories of body mass as health problems, for example, in population-based studies [74]. Third, severe obesity also represents a major problem in anesthesia and surgery. However, this study did not analyze obese patients separately as they were too few in our data (*n* = 253) and propensity matching was insufficient for rigorous analysis. Further research is required to clarify the relationship between severe obesity and PONV.

## 5. Conclusions

Multivariate analysis with PSM showed that being overweight or obese was associated with a reduced incidence of PONV compared with that of having a normal BMI but being underweight did not. Our findings indicate that a higher-than-normal BMI reduces the risk of PONV. Further prospective population-based studies are needed to determine the mechanism underlying the association between higher-than-normal BMI and reduced risk of PONV.

## Figures and Tables

**Figure 1 jcm-09-01612-f001:**
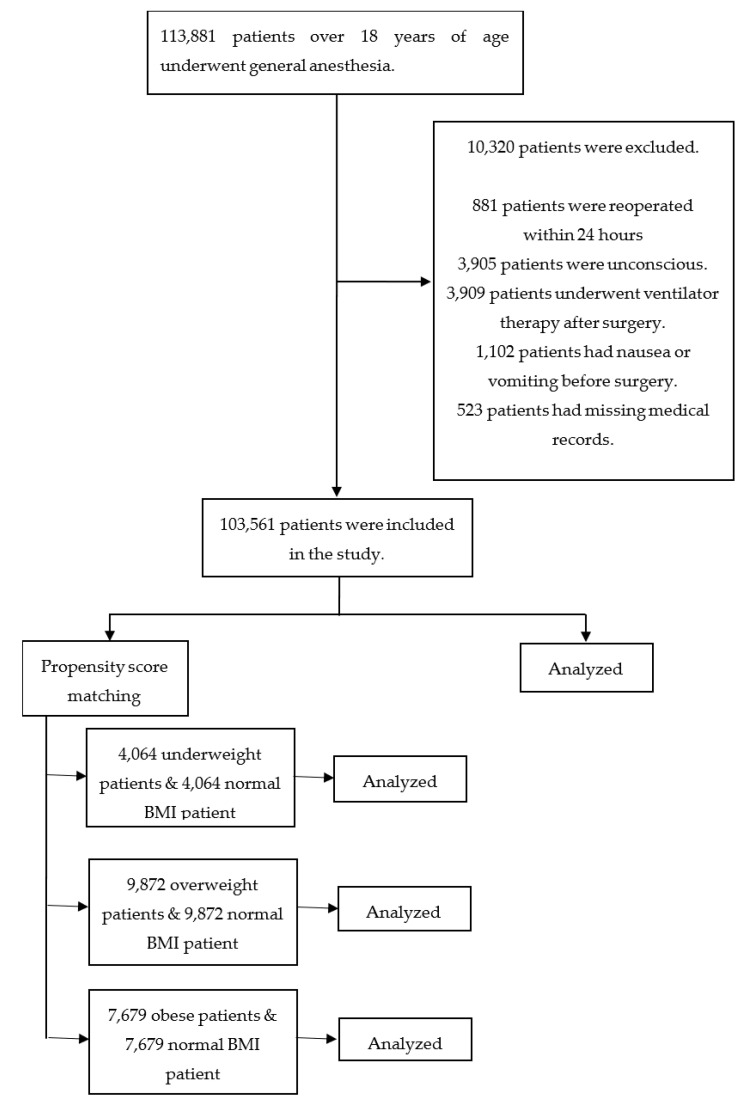
Flow chart.

**Figure 2 jcm-09-01612-f002:**
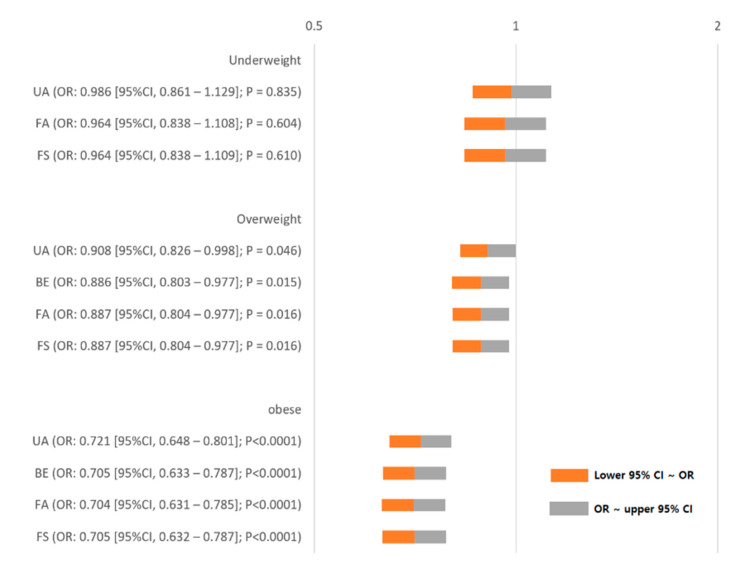
ORs for development of PONV in underweight, overweight, and obese patients. UA, unadjusted; FA, fully adjusted; FS, all covariates + propensity score adjusted; BE, backward elimination, CI, confidence interval; OR, odds ratio. Covariates for overweight BE: age, female, operation duration, American Society of Anesthesiologists physical status, use of N_2_O, patient-controlled analgesia, smoking, use of antiemetics, neostigmine, proton-pump inhibitors, or antibiotics, laparoscopic surgery, and transfusion. Covariates for obese BE: age, female, operation duration, use of N_2_O, patient-controlled analgesia, smoking, use of antiemetics, neostigmine, or proton-pump inhibitors, gastroesophageal reflux disease, and Levin tube.

**Table 1 jcm-09-01612-t001:** Baseline characteristics of the subjects.

		PONV (+) *n* = 10,404	PONV (−) *n* = 93,157	*p*
Age (years, mean ± SD)		50.7 ± 16.5	50.6 ± 17.1	0.538
Female (*n*, (%))		7561 (72.7)	45,470 (48.8)	<0.0001
Height (cm, mean ± SD)		160.5 ± 9.1	163.6 ± 9.4	<0.0001
BMI (*n*, (%))	<19.9	469 (4.5)	3595 (3.9)	<0.0001
	20–24.9	6250 (60.1)	52,573 (56.4)	
	25–29.9	3023 (29.1)	29,972 (32.2)	
	≤30	662 (6.4)	7017 (7.5)	
Operation time (h, mean ± SD)		2.5 ± 1.5	2.1 ± 1.5	<0.0001
ASA PS ≥ 3 (*n*, (%))		1540 (14.8)	14,162 (15.2)	0.280
N_2_O (*n*, (%))		1474 (14.2)	16,357 (17.6)	<0.0001
PCA (*n*, (%))		7261 (69.8)	40,869 (43.9)	<0.0001
DM (*n*, (%))		1184 (11.4)	12,274 (13.2)	<0.0001
Smoking (*n*, (%))		899 (8.6)	18,118 (19.4)	<0.0001
Opioids (*n*, (%))		9465 (91.0)	81,974 (88.0)	<0.0001
Antiemetics (*n*, (%))		7563 (72.7)	47,421 (50.9)	<0.0001
Neostigmine (*n*, (%))		3292 (31.6)	23,380 (25.1)	<0.0001
PPI (*n*, (%))		709 (6.8)	6599 (7.1)	0.309
GERD (*n*, (%))		171 (1.6)	1653 (1.8)	0.336
Antibiotics (*n*, (%))		9150 (87.9)	81,224 (87.2)	0.028
Levin tube (*n*, (%))		227 (2.2)	1743 (1.9)	0.028
Laparoscopic surgery (*n*, (%))		3189 (30.7)	21,657 (23.2)	<0.0001
Transfusion (*n*, (%))		455 (4.4)	3425 (3.7)	<0.0001

PONV, postoperative nausea and vomiting; SD, standard deviation; ASAPS, American Society of Anesthesiologists physical status; PCA, patient-controlled analgesia; DM, diabetes; GERD, gastroesophageal reflux disease; PPI, proton-pump inhibitor; BMI, body mass index.

**Table 2 jcm-09-01612-t002:** Unadjusted and fully adjusted odds ratios (ORs) of each variable for postoperative nausea and vomiting.

	Unadjusted		Fully Adjusted	
	OR (95% CI)	*p*	OR (95% CI)	*p*
Age	1.00 (1.00–1.00)	0.549	0.99 (0.99–1.00)	<0.0001
Female	2.79 (2.67–2.92)	<0.0001	2.21 (2.06–2.36)	<0.0001
Height	0.97 (0.96–0.97)	<0.0001	1.00 (0.99–1.00)	0.039
Operation time	1.14 (1.12–1.15)	<0.0001	1.06 (1.04–1.07)	<0.0001
ASA PS 3–5 vs. 1–2	0.97 (0.92–1.03)	0.280	0.88 (0.83–0.95)	<0.0001
N_2_O	0.78 (0.73–0.82)	<0.0001	0.81 (0.76–0.87)	<0.0001
PCA	2.96 (2.83–3.09)	<0.0001	2.21 (2.05–2.39)	<0.0001
Diabetes	0.85 (0.79–0.90)	<0.0001	0.93 (0.87–1.00)	0.049
Smoking	0.39 (0.37–0.42)	<0.0001	0.61 (0.56–0.65)	<0.0001
Opioids	1.38 (1.28–1.48)	<0.0001	1.10 (1.01–1.20)	0.025
Antiemetics	2.57 (2.46–2.69)	<0.0001	1.47 (1.36–1.60)	<0.0001
Neostigmine	1.38 (1.32–1.44)	<0.0001	1.40 (1.33–1.47)	<0.0001
Proton-pump inhibitor	0.96 (0.89–1.04)	0.310	0.84 (0.77–0. 92)	<0.0001
GERD	0.93 (0.79–1.08)	0.336	1.14 (0.97–1.34)	0.115
Antibiotics	1.07 (1.01–1.14)	0.028	0.88 (0.83–0.95)	0.001
Levin tube	1.17 (1.02–1.35)	0.028	0.80 (0.70–0.93)	0.003
Laparoscopic surgery	1.46 (1.40–1.53)	<0.0001	1.17 (1.12–1.23)	<0.0001
Transfusion	1.20 (1.08–1.32)	<0.0001	0.85 (0.77–0.94)	0.002
Underweight vs. normal BMI	1.10 (0.99–1.21)	0.068	1.05 (0.94–1.16)	0.406
Overweight vs. normal BMI	0.85 (0.81–0.89)	<0.0001	0.91 (0.87–0.96)	<0.0001
Obese vs. normal BMI	0.79 (0.73–0.86)	<0.0001	0.77 (0.71–0.84)	<0.0001

PONV, postoperative nausea and vomiting; OR, odds ratio; CI, confidence interval; ASAPS, American Society of Anesthesiologists physical status; PCA, patient-controlled analgesia; GERD, gastroesophageal reflux disease; PPI, proton-pump inhibitor; BMI, body mass index.

**Table 3 jcm-09-01612-t003:** Baseline characteristics of normal-BMI and underweight subjects, and comparison before and after propensity score matching.

	Before PSM				After PSM			
	Normal (*n* = 58,823)	UW (*n* = 4064)	D	*p*	Normal (*n* = 4064)	UW (*n* = 4064)	D	*p*
Age	50.3 (17.4)	49.8 (21.9)	0.02	0.147	50.1 (18.4)	49.8 (21.9)	0.01	0.598
Height	163.1 (9.0)	162.5 (8.8)	0.07	<0.0001	162.3 (8.8)	162.5 (8.8)	0.01	0.535
Female	31,555 (53.6)	2498 (61.5)	0.18	<0.0001	2501 (61.5)	2498 (61.5)	0.03	0.945
Operation time	2.1 (1.5)	2.1 (1.4)	0.01	0.630	2.1(1.4)	2.1 (1.4)	0.04	0.108
ASA PS ≥ 3	8735 (14.8)	953 (23.4)	0.31	<0.0001	983 (24.2)	953 (23.4)	0.02	0.435
N_2_O	10,127 (17.2)	699 (17.2)	<0.01	0.797	749 (18.4)	699 (17.2)	0.05	0.147
PCA	27,337 (46.5)	2002 (49.3)	0.06	0.001	1942 (48.2)	2002 (49.3)	0.02	0.340
Diabetes	6567 (11.2)	417 (10.3)	0.05	0.076	423 (10.4)	417 (10.3)	0.01	0.827
Smoking	10,500(17.9)	781 (19.2)	0.05	0.028	796 (19.6)	781 (19.2)	0.03	0.293
Opioids	51,867 (88.2)	3521 (86.6)	0.08	0.003	3480 (85.6)	3521 (86.6)	0.05	0.188
Antiemetics	31,178 (53.0)	2288 (56.3)	0.07	<0.0001	2272 (55.9)	2288 (56.3)	0.01	0.721
Neostigmine	15,374 (26.1)	1059 (26.1)	<0.01	0.913	1023 (25.2)	1059 (26.1)	0.03	0.360
PPI	4268 (7.3)	406 (10.0)	0.19	<0.0001	442 (10.9)	406 (10.0)	0.05	0.191
GERD	975 (1.7)	63 (1.6)	0.04	0.604	65 (1.6)	63 (1.6)	0.02	0.859
Antibiotics	51,372 (87.3)	3533 (86.9)	0.02	0.460	3527 (86.8)	3533 (86.9)	0.01	0.844
Levin tube	1184 (2.0)	112 (2.8)	0.18	0.001	99 (2.4)	112 (2.8)	0.07	0.365
Laparoscopic surgery	14,692 (25.0)	990 (24.4)	0.02	0.380	986 (24.3)	990 (24.4)	<0.01	0.918
Transfusion	2380 (4.0)	327 (8.0)	0.40	<0.0001	291 (7.2)	327 (8.0)	0.07	0.132

BMI, body mass index; PSM, propensity score matching; UW, underweight; D, standard mean difference; ASA PS, American Society of Anesthesiologists physical status; PCA, patient-controlled analgesia; PPI, proton-pump inhibitor; GERD, gastroesophageal reflux disease.Type of Surgery other than laparoscopic surgery before propensity score matching: Normal (brain surgery, 479; eye surgery, 1372; pharynx and tonsil surgery, 1361; ear surgery, 1030) vs. UW (brain surgery, 48; eye surgery, 96; pharynx and tonsil surgery, 76; ear surgery, 44).Type of Surgery other than laparoscopic surgery after propensity score matching: Normal (brain surgery, 42; eye surgery, 78; pharynx and tonsil surgery, 108; ear surgery, 50) vs. UW (brain surgery, 48; eye surgery, 96; pharynx and tonsil surgery, 76; ear surgery, 44).

**Table 4 jcm-09-01612-t004:** Baseline characteristics of normal-BMI and overweight subjects, and comparison before and after propensity score matching.

	Before PSM				After PSM			
	Normal (*n* = 58,823)	OW (*n* = 32,995)	D	*p*	Normal (*n* = 9872)	OW (*n* = 9872)	D	*p*
Age	50.3 (17.4)	52.1 (15.7)	0.11	<0.0001	51.0 (17.1)	50.6 (15.5)	0.02	0.159
Height	163.1 (9.0)	163.6 (9.8)	0.05	<0.0001	163.5 (9.6)	163.4 (10.1)	0.01	0.512
Female	31,555 (53.6)	14,862 (45.0)	0.19	<0.0001	4806 (48.7)	4939 (50.0)	0.03	0.058
Operation time	2.1 (1.5)	2.2 (1.5)	0.04	<0.0001	2.1 (1.4)	2.1 (1.4)	0.01	0.554
ASA PS ≥ 3	8735 (14.8)	4847 (14.7)	0.01	0.514	1419 (14.4)	1370 (13.9)	0.02	0.317
N_2_O	10,127 (17.2)	5739 (17.4)	0.01	0.797	1801 (18.2)	1728 (17.5)	0.03	0.175
PCA	27,337 (46.5)	15,281 (46.3)	<0.01	0.001	4512 (45.7)	4491 (45.5)	<0.01	0.764
Diabetes	6567 (11.2)	5049 (15.3)	0.20	<0.0001	889 (9.0)	998 (10.1)	0.07	0.008
Smoking	10,500 (17.9)	6243 (18.9)	0.04	<0.0001	1965 (19.9)	1865 (18.9)	0.04	0.072
Opioids	51,867 (88.2)	29,277 (88.7)	0.03	0.012	8776 (88.9)	8772 (88.9)	<0.01	0.928
Antiemetics	31,178 (53.0)	17,474 (53.0)	<0.01	0.899	17,525 (53.1)	17,474 (53.0)	0.02	0.332
Neostigmine	15,374 (26.1)	8411 (25.5)	0.02	0.033	2693 (25.2)	2585 (25.5)	0.04	0.082
PPI	4268 (7.3)	2215 (6.7)	0.05	0.002	667 (6.8)	182 (6.9)	0.01	0.672
GERD	975 (1.7)	655 (2.0)	0.10	<0.0001	165 (1.9)	149 (2.0)	0.06	0.363
Antibiotics	51,372 (87.3)	28,781 (86.9)	0.01	0.647	8647 (87.6)	8632 (86.9)	0.01	0.747
Levin tube	1184 (2.0)	579 (1.8)	0.08	0.006	175 (1.8)	146 (1.5)	0.10	0.103
Laparoscopic surgery	14,692 (25.0)	7446 (22.6)	0.07	<0.0001	2180 (22.1)	2277 (23.1)	0.03	0.099
Transfusion	2380 (4.0)	915 (2.8)	0.22	<0.0001	232 (2.4)	184 (1.9)	0.13	0.230

BMI, body mass index; PSM, propensity score matching; OW, overweight; D, standard mean difference; ASA PS, American society of Anesthesiologists physical status; PCA, patient-controlled analgesia; GERD, gastroesophageal reflux disease.Type of Surgery other than laparoscopic surgery before propensity score matching: Normal (brain surgery, 479; eye surgery, 1372; pharynx and tonsil surgery, 1361; ear surgery, 1030) vs. OW (brain surgery, 258; eye surgery, 830; pharynx and tonsil surgery, 817; ear surgery, 657).Type of Surgery other than laparoscopic surgery after propensity score matching: Normal (brain surgery, 70; eye surgery, 237; pharynx and tonsil surgery, 225; ear surgery, 159) vs. OW (brain surgery, 75; eye surgery, 235; pharynx and tonsil surgery, 292; ear surgery, 196).

**Table 5 jcm-09-01612-t005:** Baseline characteristics of normal-BMI and obese subjects, and comparison before and after propensity score matching.

	Before PSM				After PSM			
	Normal (*n* = 58,823)	Obese (*n* = 7679)	D	*p*	Normal (*n* = 7679)	Obese (*n* = 7679)	D	*p*
Age	50.3 (17.4)	46.4 (16.1)	0.02	0.147	46.5 (16.8)	46.4 (16.1)	0.01	0.670
Height	163.1 (9.0)	163.7 (10.4)	0.06	<0.0001	163.5 (9.0)	163.7 (10.4)	0.01	0.598
Female	31,555 (53.6)	4119 (53.6)	0.02	0.994	4114 (53.6)	4119 (53.6)	<0.01	0.936
Operation time	2.1 (1.5)	2.2 (1.5)	0.04	<0.0001	2.1 (1.4)	2.2 (1.5)	<0.01	0.988
ASA PS ≥ 3	8735 (14.8)	1167 (15.2)	0.31	0.421	1188 (15.5)	1167 (15.2)	0.01	0.638
N_2_O	10,127 (17.2)	1266 (16.5)	0.03	0.111	1301 (16.9)	1266 (16.5)	0.02	0.449
PCA	27,337 (46.5)	3510 (45.7)	0.02	0.207	3562 (46.4)	3510 (45.7)	0.02	0.400
Diabetes	6567 (11.2)	1425 (18.6)	0.33	<0.0001	1422 (18.5)	1425 (18.6)	<0.01	0.950
Smoking	10,500 (17.9)	1493 (19.4)	0.06	0.001	1485 (19.3)	1493 (19.4)	<0.01	0.870
Opioids	51,867 (88.2)	6774 (88.2)	<0.01	0.919	6775 (85.6)	6774 (88.2)	<0.01	0.980
Antiemetics	31,178 (53.0)	4044 (52.7)	0.01	0.575	4110 (53.5)	4044 (52.7)	0.02	0.286
Neostigmine	15,374 (26.1)	1828 (23.8)	0.07	<0.0001	1784 (25.2)	1828 (23.8)	0.02	0.403
PPI	4268 (7.3)	419 (5.5)	0.17	<0.0001	442 (10.9)	419 (5.5)	<0.01	0.943
GERD	975 (1.7)	131 (1.7)	0.02	0.755	144 (1.9)	131 (1.7)	0.05	0.429
Antibiotics	51,372 (87.3)	6688 (87.1)	0.01	0.555	6659 (86.7)	6688 (87.1)	0.02	0.488
Levin tube	1184 (2.0)	95 (1.2)	0.27	<0.0001	97 (1.3)	95 (1.2)	0.01	0.885
Laparoscopic surgery	14,692 (25.0)	1718 (22.4)	0.08	<0.0001	1709 (22.3)	1718 (22.4)	<0.01	0.862
Transfusion	2380 (4.0)	258 (3.4)	0.11	0.004	253 (3.3)	258 (3.4)	0.01	0.822

BMI, body mass index; PSM, propensity score matching; D, standard mean difference; ASA PS, American Society of Anesthesiologists physical status; PCA, patient-controlled analgesia; GERD, gastroesophageal reflux disease.Type of Surgery other than laparoscopic surgery before propensity score matching: Normal (brain surgery, 479; eye surgery, 1372; pharynx and tonsil surgery, 1361; ear surgery, 1030) vs. Obese (brain surgery, 56; eye surgery, 170; pharynx and tonsil surgery, 283; ear surgery, 107).Type of Surgery other than laparoscopic surgery after propensity score matching: Normal (brain surgery, 82; eye surgery, 190; pharynx and tonsil surgery, 221; ear surgery, 132) vs. Obese (brain surgery, 56; eye surgery, 170; pharynx and tonsil surgery, 283; ear surgery, 107).

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
