# Peer review of "Effect of Body Mass Index on Postoperative Nausea and Vomiting: Propensity Analysis"

_jcm, 2020, doi:10.3390/jcm9061612_

Round 1
Reviewer 1 Report
The study on Effect of Body Mass Index on Postoperative Nausea and Vomiting: Propensity analysis by JJ Lee et al., is interesting. However, the discussion is too light and can be improved.
Introduction: L. 34-35, it seems likely that other groups (reference no. 12-14) have studied or reported on BMI-PONV. Thus, authors must state strongly how the present study is different from the previous ones or how this present study will add new information to the field. The authors might already state that they performed propensity score matching in this study. For readers outside the field to understand, the specialty of "propensity score matching" should be explained a bit more --> why it will make this study different from the previous studies.
Methods: A short definition of the odds ratio (OR) should be provided, what OR tells? A correlation?
Discussion: 1. The study itself is interesting. It may lead to therapeutic approaches in the future. However, this manuscript is like a report. Although the authors already stated in the conclusion that further studies are needed to clarify the underlying mechanisms, the possible mechanisms, which I believed there should be papers saying about this out there, should be proposed or hypothesized. If there is no paper proposing the possible mechanisms, the authors should at least gather the information and give the hypothesis.
2. The results in the present study are opposite with the general belief of most anesthesiologists who are the experts in the field and consider that an increase in BMI increases the incidence of PONV (L. 175). The authors should find a stronger explanation for this. There is no need for saying which is right or wrong, but the explanation should be provided.
Author Response
Response to Reviewer 1 Comments
Point 1: Introduction: L. 34-35, it seems likely that other groups (reference no. 12-14) have studied or reported on BMI-PONV. Thus, authors must state strongly how the present study is different from the previous ones or how this present study will add new information to the field. The authors might already state that they performed propensity score matching in this study. For readers outside the field to understand, the specialty of "propensity score matching" should be explained a bit more --> why it will make this study different from the previous studies.
Response 1: Thank you very much for your opinion. We added the specialty of "propensity core matching" to introduction part. Many PONV studies are retrospective observations studies. Because observational studies conduct research on specific groups without random assignment, they have a slightly more realistic and general advantage. It can also be controlled so that estimated therapeutic effects can be applied to clinicians, and statistical controls can be applied for specific confounding variables. However, observational studies cannot avoid selective biases in the selection of subjects. It is fundamentally impossible to deduce the cause of a phenomenon because it is not based on random assignment. The classic method used in observational studies to reduce selection bias is matching method. Propensity score matching (PSM) is used not only in observational studies but also as a way to reduce selection bias in retrospective research methods where it is difficult to apply random assignment.
L.37
Point 2: Methods: A short definition of the odds ratio (OR) should be provided, what OR tells? A correlation?
Response 2: Thank you very much for your opinion. We added "odds ratio" to introduction part.An odds ratio (OR) is a measure of association between an exposure and an outcome, and the OR in this study represents the odds that an PONV will occur given a particular exposure (underweight, overweight and obese), compared to the odds of the PONV occurring in the absence of that exposure.
L.77
Point 3: Discussion: 1. The study itself is interesting. It may lead to therapeutic approaches in the future. However, this manuscript is like a report. Although the authors already stated in the conclusion that further studies are needed to clarify the underlying mechanisms, the possible mechanisms, which I believed there should be papers saying about this out there, should be proposed or hypothesized. If there is no paper proposing the possible mechanisms, the authors should at least gather the information and give the hypothesis.
Response 3: Thank you very much for your opinion. We hypothesized a possible mechanism. No mechanism has yet been reported to reduce PONV with higher than normal BMI. Histamine, a neurotransmitter that stimulates various pathways and receptors that enable nausea and vomiting, may be associated with the mechanism that a higher BMI than normal reduces PONV. Leptin is an adipocyte-derived hormone that suppresses appetite and increase energy expenditure and activation of histaminergic system in hypothalamus and controls body weight. Amount of body fat is directly correlated circulating leptin and serum leptin increase in obese individuals and drop during weight loss or fasting. The amount that is reduced during the fasting period is significantly reduced compared to normal people. The decrease in the activity of the histamine activation system may be greater in higher BMI than normal during perioperative fasting.
L.249
Point 4: The results in the present study are opposite with the general belief of most anesthesiologists who are the experts in the field and consider that an increase in BMI increases the incidence of PONV (L. 175). The authors should find a stronger explanation for this. There is no need for saying which is right or wrong, but the explanation should be provided.
Response 4: Thank you very much for your opinion. We explained reason of belief that an increase in BMI increases the incidence of PONV. The longer the anesthesia time, the longer the reduction time of the anesthetic in obese patients, and when a hydrophilic anesthetic is administered based on TBW, the action time of the anesthetic is increased when a hydrophilic anesthetic is administered based on TBW. In addition, obese patients have high incidence of diabetes and gastroesophageal reflux disease and abdominal pressure.
L.213
Reviewer 2 Report
I enjoyed the statistics of this paper, very interesting. Jae Jun Lee et al. investigated the effects of BMI on PONV, considering other PONV risk factors. They analyzed adults over the age of 18 years who received general anesthesia, using propensity score matching. Before propensity score matching, odds ratios [ORs] for PONV were lower for overweight, P < 0.0001) or obese patients, P < 0.0001) than for normal-BMI patients. After matching, the ORs for PONV of overweight, P = 0.016) and obese patients, P < 0.0001) were low. Interestingly, the ORs of underweight patients did not differ from those of normal-BMI patients, irrespective of matching.Therefore, the incidence of PONV may be lower among adults with a higher than normal BMI.
The study concluded a multivariate analysis with propensity score matching showed that being 225 overweight or obese was associated with a reduced incidence of PONV compared with having a 226 normal BMI. These findings indicate that a relatively high BMI reduces the risk of PONV. The authors admit that further 227 prospective population-based studies are needed to explain the mechanism underlying the 228 association between a high BMI and a decreased risk of PONV.
Author Response
Response to Reviewer 2 Comments
Point 1: I enjoyed the statistics of this paper, very interesting. Jae Jun Lee et al. investigated the effects of BMI on PONV, considering other PONV risk factors. They analyzed adults over the age of 18 years who received general anesthesia, using propensity score matching. Before propensity score matching, odds ratios [ORs] for PONV were lower for overweight, P < 0.0001) or obese patients, P < 0.0001) than for normal-BMI patients. After matching, the ORs for PONV of overweight, P = 0.016) and obese patients, P < 0.0001) were low. Interestingly, the ORs of underweight patients did not differ from those of normal-BMI patients, irrespective of matching.Therefore, the incidence of PONV may be lower among adults with a higher than normal BMI.
Response 2: Thank you very much for your opinion. We changed the conclusion according to your advice.
In this study, a multivariate analysis with propensity score matching showed that being overweight or obese was associated with a reduced incidence of PONV compared with having a normal BMI, but being underweight was not significant different from normal BMI. Our findings indicate that a higher BMI than normal reduces the risk of PONV. Further prospective population-based studies are needed to explain the mechanism underlying the association between a higher BMI than normal and a decreased risk of PONV.
L.285
Reviewer 3 Report
This study by Lee et al. assesses the Impact of BMI on postoperative vomiting and nausea. The large number of cases makes the data very convincing however there are two issues that need to be adressed: what anestetic agents did the patients receive and at what dosage.
Furthermore the potential pathophysiologic mechanisms leading to more PONV in normal BMI patients is not sufficiently discussed.
Author Response
Response to Reviewer 3 Comments
Point 1: This study by Lee et al. assesses the Impact of BMI on postoperative vomiting and nausea. The large number of cases makes the data very convincing however there are two issues that need to be adressed: what anestetic agents did the patients receive and at what dosage.
Response 1: Thanks for your comments. Your advice is a very important issue. Anesthetics vary widely, and in general, intravenous anesthetics are used to induce anesthesia and inhaled anesthetics are used to maintain anesthesia, but several kinds of anesthetics are often used in the anesthesia process. Inhalation anesthesia is the most commonly used anesthetic. The inhalation anesthetic is controlled to the concentration, and the concentration changes from time to time in response to changes in vital signs during surgery, and the amount of anesthetic consumed varies depending on the fresh gas flow provided to the patient. We can know the concentration of inhalation anesthesia during surgery but can’t know amount used. In the case of intravenous anesthetics, we can know the concentration and infusion rate, but we can’t know total amount used because it is common for the infusion rate to change several times during surgery. Our study included N2O, a well-known risk factor for PONV among inhalation anesthetics, as a covariate. N2O is often used as a supplemental anesthetic for other main inhalation or intravenous anesthetics.
Reviewer 4 Report
Dear Authors/Editors
This is an excellent description of a multivariable analysis that includes a propensity analysis to determine if obesity is a risk factor for PONV. Importanly, it may address weight bias among health care professionals in showing a decreased risk of PONV in people with obesity. I do have a few comments on how to improve the paper that I want to share with the authors.
A discussion on how the data were collected and what consitituted PONV when collecting the data would would add to the quality of the paper. We know that postoperative nausea and postoperative vomiting should be measured and analysed separately and then together, please provide rationale for only combining them as PONV in the analysis.
PONV varies across surgical sites/procedures (e.g. high posterior fossa neurosurgery, some ENT procedures, etc.), please discuss why you chose not to control for, or complete a subgroup analysis, for surgery type in your analysis. Type of surgery could be included in your descriptive table even if not part of the analysis.
Importantly, severe obesity (e.g. BMI greater than 40 or greater than 35 if a person has obesity related health conditions OR obesity categorised using of an obesity staging system such as the Edmonton Obesity Staging System - EOSS); is an essential aspect to consider in any research that focuses on obesity. Please identify how many people with severe obesity were included the study and discuss the rationale for not including this category in your analysis. Alternative, a separate analysis looking at PONV and severe obesity would enhance the study.
Author Response
Response to Reviewer 4 Comments
Point 1: A discussion on how the data were collected and what consitituted PONV when collecting the data would add to the quality of the paper. We know that postoperative nausea and postoperative vomiting should be measured and analysed separately and then together, please provide rationale for only combining them as PONV in the analysis.
Response 1: Thank you very much for your opinion. We defined PONV as a case where postoperative nausea and vomiting were recorded on the medical record. We do not have considered nausea or vomiting events as separate outcomes in this study. Although there are pathophysiological differences between the two, many studies have reported the similar risk factors for nausea and vomiting after surgery. The postoperative nausea be considered a potential symptom of vomiting, and the accompanying occurrence of nausea and vomiting is more than the individual occurrence of nausea and vomiting. Many scoring systems to predict PONV also do not predict postoperative nausea and vomiting separately. The medicines to treat postoperative nausea or vomiting are similar.
L.63
Point 2: PONV varies across surgical sites/procedures (e.g. high posterior fossa neurosurgery, some ENT procedures, etc.), please discuss why you chose not to control for, or complete a subgroup analysis, for surgery type in your analysis. Type of surgery could be included in your descriptive table even if not part of the analysis.
Response 2: Thank you very much for your opinion. Studies of the effect of the type of surgery on the incidence of PONV have reported conflicting results. A systemic review of PONV suggests that laparoscopic surgery is associated with an increased risk of PONV. In this study, we included laparoscpic surgery as a covariate on incidence of PONV. Strabismus surgery, adenotonsillectomy, inguinal scrotal and penile procedures are reported as risk factors for PONV. However, we did not analyze the surgeries separately because they were a study conducted in children and, in general, the patients targeted for the surgeries were most children. In each table, the number of other types of surgery other than laparoscopic surgery is described.
L.258
Point 3: Importantly, severe obesity (e.g. BMI greater than 40 or greater than 35 if a person has obesity related health conditions OR obesity categorised using of an obesity staging system such as the Edmonton Obesity Staging System - EOSS); is an essential aspect to consider in any research that focuses on obesity. Please identify how many people with severe obesity were included the study and discuss the rationale for not including this category in your analysis. Alternative, a separate analysis looking at PONV and severe obesity would enhance the study.
Response 3: Thank you very much for your opinion. We felt it was very good advice. We can analyze patients with BMI of 40 or higher with our data. However, very few Koreans have BMI of 40 or more. In our data, BMI 40 has only 254 patients, and no satisfactory propensity match has been made for analysis. We will conduct a study on patients with BMI or higher if sufficient data is collected later.
L.280